# Fast and Flexible Monotonic Functions with Ensembles of Lattices

**K. Canini, A. Cotter, M. R. Gupta, M. Milani Fard, J. Pfeifer**
Google Inc.
1600 Amphitheatre Parkway, Mountain View, CA 94043
{canini,acotter,mayagupta,janpf,mmilanifard}@google.com

## Abstract

For many machine learning problems, there are some inputs that are known to be positively (or negatively) related to the output, and in such cases training the model to respect that monotonic relationship can provide regularization, and makes the model more interpretable. However, flexible monotonic functions are computationally challenging to learn beyond a few features. We break through this barrier by learning ensembles of monotonic calibrated interpolated look-up tables (lattices). A key contribution is an automated algorithm for selecting feature subsets for the ensemble base models. We demonstrate that compared to random forests, these ensembles produce similar or better accuracy, while providing guaranteed monotonicity consistent with prior knowledge, smaller model size and faster evaluation.

## 1 Introduction

A long-standing challenge in machine learning is to learn flexible *monotonic* functions [1] for classification, regression, and ranking problems. For example, all other features held constant, one would expect the prediction of a house's cost to be an increasing function of the size of the property. A regression trained on noisy examples and many features might not respect this simple monotonic relationship everywhere, due to overfitting. Failing to capture such simple relationships is confusing for users. Guaranteeing monotonicity enables users to trust that the model will behave reasonably and predictably in all cases, and enables them to understand at a high-level how the model responds to monotonic inputs [2]. Prior knowledge about monotonic relationships can also be an effective regularizer [3].

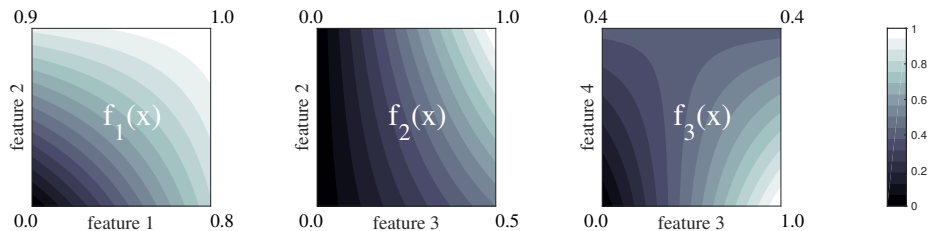

Figure 1: Contour plots for an ensemble of three lattices, each of which is a linearly-interpolated look-up table. The first lattice $f_1$ acts on features 1 and 2, the second lattice $f_2$ acts on features 2 and 3, and $f_3$ acts on features 3 and 4. Each $2 \times 2$ look-up table has four parameters: for example, for $f_1(x)$ the parameters are $\theta_1 = [0, 0.9, 0.8, 1]$. Each look-up table parameter is the function value for an extreme input, for example, $f_1([0, 1]) = \theta_1[2] = 0.9$. The ensemble function $f_1 + f_2 + f_3$ is monotonic with respect to features 1, 2 and 3, but not feature 4.

Table 1: Key notation used in the paper.

| Symbol | Definition |
| --- | --- |
| $D$ | number of features |
| $\mathcal{D}$ | set of features $1, 2, \ldots, D$ |
| $S$ | number of features in each lattice |
| $L$ | number of lattices in the ensemble |
| $s_\ell \subset \mathcal{D}$ | $\ell$th lattice's set of $S$ feature indices |
| $(x_i, y_i) \in [0,1]^D \times \mathbb{R}$ | $i$th training example |
| $x[s_\ell] \in [0,1]^S$ | $x$ for feature subset $s_\ell$ |
| $\Phi(x) : [0,1]^D \to [0,1]^{2^D}$ | linear interpolation weights for $x$ |
| $v \in \{0,1\}^D$ | a vertex of a $2^D$ lattice |
| $\theta \in \mathbb{R}^{2^D}$ | parameter values for a $2^D$ lattice |

Simple monotonic functions can be learned with a linear function forced to have positive coefficients, but learning flexible monotonic functions is challenging. For example, multi-dimensional isotonic regression has complexity $O(n^4)$ for $n$ training examples [4]. Other prior work learned monotonic functions on small datasets and very low-dimensional problems; see [2] for a survey. The largest datasets used in recent papers on training monotonic neural nets included two features and 3434 examples [3], one feature and 30 examples [5], and six features and 174 examples [6]. Another approach that uses an ensemble of rules on a modified training set, scales poorly with the dataset size and does not provide monotonicity guarantees on test samples [7].

Recently, *monotonic lattice regression* was proposed [2], which extended lattice regression [8] to learn a monotonic interpolated look-up table by adding linear inequality constraints to constrain adjacent look-up table parameters to be monotonic. See Figure 1 for an illustration of three such lattice functions. Experiments on real-world problems with millions of training examples showed similar accuracy to random forests [9], which generally perform well [10], but with the benefit of guaranteed monotonicity. Monotonic functions on up to sixteen features were demonstrated, but that approach is fundamentally limited in its ability to scale to higher input dimensions, as the number of parameters for a lattice model scales as $O(2^D)$.

In this paper, we break through previous barriers on $D$ by proposing strategies to learn a monotonic ensemble of lattices. The main contributions of this paper are: (i) proposing different architectures for ensembles of lattices with different trade-offs for flexibility, regularization and speed, (ii) theorem showing lattices can be merged, (iii) an algorithm to automatically learn feature subsets for the base models, (iv) extensive experimental analysis on real data showing results that are similar to or better than random forests, while respecting prior knowledge about monotonic features.

## 2 Ensemble of Lattices

Consider the usual supervised machine learning set-up of training sample pairs $\{(x_i, y_i)\}$ for $i = 1, \ldots, n$. The label is either a real-valued $y_i \in \mathcal{R}$, or a binary classification label $y_i \in \{-1, 1\}$. We assume that sensible upper and lower bounds can be set for each feature, and without loss of generality, the feature is then scaled so that $x_i \in [0,1]^D$. Key notation is summarized in Table 1. We propose learning a weighted ensemble of $L$ lattices:

$$F(x) = \alpha_0 + \sum_{\ell=1}^{L} \alpha_\ell f(x[s_\ell]; \theta_\ell), \qquad (1)$$

where each lattice $f(x[s_\ell]; \theta_\ell)$ is a lattice function defined on a subset of features $s_\ell \subset \mathcal{D}$, and $x[s_\ell]$ denotes the $S \times 1$ vector with the components of $x$ corresponding to the feature set $s_\ell$. We require each lattice to have $S$ features, i.e. $|s_l| = S$ and $\theta_l \in \mathbb{R}^{2^S}$ for all $l$. The $\ell$th lattice is a linearly interpolated look-up table defined on the vertices of the $S$-dimensional unit hypercube:

$$f(x[s_\ell]; \theta_\ell) = \theta_\ell^T \Phi(x[s_\ell]),$$

where $\theta_\ell \in \mathbb{R}^{2^S}$ are the look-up table parameters, and $\Phi(x[s]) : [0,1]^S \to [0,1]^{2^S}$ are the linear interpolation weights for $x$ in the $\ell$th lattice. See Appendix A for a review of multilinear and simplex interpolations.

## 2.1 Monotonic Ensemble of Lattices

Define a function $f$ to be *monotonic* with respect to feature $d$ if $f(x) \geq f(z)$ for any two feature vectors $x, z \in \mathbb{R}^D$ that satisfy $x[d] \geq z[d]$ and $x[m] = z[m]$ for $m \neq d$. A lattice function is monotonic with respect to a feature if the look-up table parameters are nondecreasing as that feature increases, and a lattice can be constrained to be monotonic with an appropriate set of sparse linear inequality constraints [2].

To guarantee that an ensemble of lattices is monotonic with respect to feature $d$, it is sufficient that each lattice be monotonic in feature $d$ and that we have positive weights $\alpha_\ell \geq 0$ for all $\ell$, by linearity of the sum over lattices in (1). Constraining each base model to be monotonic provides only a subset of the possible monotonic ensemble functions, as there could exist monotonic functions of the form (1) that are not monotonic in each $f_\ell(x)$.

In this paper, for notational simplicity and because we find it the most empirically useful case for machine learning, we only consider lattices that have two vertices along each dimension, so a look-up table on $S$ features has $2^S$ parameters. Generalization to larger look-up tables is trivial [11, 2].

## 2.2 Ensemble of Lattices Can Be Merged into One Lattice

We show that an ensemble of lattices as expressed by (1) is equivalent to a single lattice defined on all $D$ features. This result is important for two practical reasons. First, it shows that training an ensemble over subsets of features is a regularized form of training one lattice with all $D$ features. Second, this result can be used to merge small lattices that share many features into larger lattices on the superset of their features, which can be useful to reduce evaluation time and memory use of the ensemble model.

**Theorem 1** *Let $F(x) := \sum_{l=1}^{L} \alpha_l \theta_l^T \Phi(x[s_l])$ as described in (1) where $\Phi$ are either multilinear or simplex interpolation weights. Then there exists a $\theta \in \mathbb{R}^{2^D}$ such that $F(x) = \theta^T \Phi(x)$ for all $x \in [0, 1]^D$.*

The proof and an illustration is given in the supplementary material.

# 3 Feature Subset Selection for the Base Models

A key problem is which features should be combined in each lattice. The random subspace method [12] and many variants of random forests randomly sample subsets of features for the base models. Sampling the features to increase the likelihood of selecting combinations of features that better correlate with the label can produce better random forests [13]. Ye et al. first estimated the informativeness of each feature by fitting a linear model and dividing the features up into two groups based on the linear coefficients, then randomly sampled the features considered for each tree from both groups, to ensure strongly informative features occur in each tree [14]. Others take the random subset of features for each tree as a starting point, but then create more diverse trees. For example, one can increase the diversity of the features in a tree ensemble by changing the splitting criterion [15], or by maximizing the entropy of the joint distribution over the leaves [16].

We also consider randomly selecting the subset of $S$ features for each lattice uniformly and independently without replacement from the set of $D$ features, we refer to this as a *random tiny lattice* (RTL). To improve the accuracy of ensemble selection, we can draw $K$ independent RTL ensembles using $K$ different random seeds, train each ensemble independently, and select the one with the lowest training or validation or cross-validation error. This approach treats the random seed that generates the RTL as a hyper-parameter to optimize, and in the extreme of sufficiently large $K$, this strategy can be used to minimize the empirical risk. The computation scales linearly in $K$, but the $K$ training jobs can be parallelized.

## 3.1 Crystals: An Algorithm for Selecting Feature Subsets

As a more efficient alternative to randomly selecting feature subsets, we propose an algorithm we term *Crystals* to jointly optimize the selection of the $L$ feature subsets. Note that linear interactions between features can be captured by the ensemble's linear combination of base models, but nonlinear

interactions must be captured by the features occurring together in a base model. This motivates us to propose choosing the feature subsets for each base model based on the importance of pairwise nonlinear interactions between the features.

To measure pairwise feature interactions, we first separately (and in parallel) train lattices on all possible pairs of features, that is $L = \binom{D}{2}$ lattices, each with $S = 2$ features. Then, we measure the nonlinearity of the interaction of any two features $d$ and $\tilde{d}$ by the *torsion* of their lattice, which is the squared difference between the slopes of the lattice's parallel edges [2]. Let $\theta_{d,\tilde{d}}$ denote the parameters of the $2\times2$ lattice for features $d$ and $\tilde{d}$. The torsion of the pair $(d,\tilde{d})$ is defined as:

$$\tau_{d,\tilde{d}} \stackrel{\text{def}}{=} \left( (\theta_{d,\tilde{d}}[1] - \theta_{d,\tilde{d}}[0]) - (\theta_{d,\tilde{d}}[3] - \theta_{d,\tilde{d}}[2]) \right)^2 . \tag{2}$$

If the lattice trained on features $d$ and $\tilde{d}$ is just a linear function, its torsion is $\tau_{d,\tilde{d}} = 0$, whereas a large torsion value implies a nonlinear interaction between the features. Given the torsions of all pairs of features, we propose using the $L$ feature subsets $\{s_\ell\}$ that maximize the weighted total pairwise torsion of the ensemble:

$$H(\{s_\ell\}) \stackrel{\text{def}}{=} \sum_{\ell=1}^{L} \sum_{\substack{d,\tilde{d}\in s_\ell \\ d\neq\tilde{d}}} \tau_{d,\tilde{d}} \, \gamma^{\sum_{\ell'=1}^{\ell-1} I(d,\tilde{d}\in s_{\ell'})}, \tag{3}$$

where $I$ is an indicator. The discount value $\gamma$ is a hyperparameter that controls how much the value of a pair of features diminishes when repeated in multiple lattices. The extreme case $\gamma = 1$ makes the objective (3) optimized by the degenerate case that all $L$ lattices include the same feature pairs that have the highest torsion. The other extreme of $\gamma = 0$ only counts the first inclusion of a pair of features in the ensemble towards the objective, and results in unnecessarily diverse lattices. We found a default of $\gamma = \frac{1}{2}$ generally produces good, diverse lattices, but $\gamma$ can also be optimized as a hyperparameter.

In order to select the subsets $\{s_\ell\}$, we first choose the number of times each feature is going to be used in the ensemble. We make sure each feature is used at least once and then assign the rest of the feature counts proportional to the median of each feature's torsion with other features. We initialize a random ensemble that satisfies the selected feature counts and then try to maximize the objective in (3) using a greedy swapping method: We loop over all pairs of lattices, and swap any features between them that increase $H(\{s_\ell\})$, until no swaps improve the objective. This optimization takes a small fraction of the total training time for the ensemble. One can potentially improve the objective by using a stochastic annealing procedure, but we find our deterministic method already yields good solutions in practice.

## 4   Calibrating the Features

Accuracy can be increased by calibrating each feature with a one-dimensional monotonic piecewise linear function (PLF) before it is combined with other features [17, 18, 2]. These calibration functions can approximate log, sigmoidal, and other useful feature pre-processing transformations, and can be trained as part of the model.

For an ensemble, we consider two options. Either a set of $D$ calibrators *shared* across the base models (one PLF per feature), or $L$ sets of calibrators, one set of $S$ calibrators for each base model, for a total of $LS$ calibrators. Use of separate calibrators provides more flexibility, but increases the potential for overfitting, increases evaluation time, and removes the ability to merge lattices.

Let $c(x[s_\ell]; \nu_\ell) : [0,1]^S \to [0,1]^S$ denote the vector-valued calibration function on the feature subset $s_\ell$ with calibration parameters $\nu_\ell$. The ensemble function will thus be:

$$\text{Separate Calibration: } F(x) = \alpha_0 + \sum_{\ell=1}^{L} \alpha_\ell f(c(x[s_\ell]; \nu_\ell); \theta_\ell) \tag{4}$$

$$\text{Shared Calibration: } F(x) = \alpha_0 + \sum_{\ell=1}^{L} \alpha_\ell f(c(x[s_\ell]; \nu[s_\ell]); \theta_\ell), \tag{5}$$

where $\nu$ is the set of all shared calibration parameters and $\nu[s_\ell]$ is the subset corresponding to the feature set $s_\ell$. Note that Theorem 1 holds for shared calibrators, but does not hold for separate calibrators.

We implement these PLF's as in [2]. The PLF's are monotonic if the adjacent parameters in each PLF are monotonic, which can be enforced with additional linear inequality constraints [2]. By composition, if the calibration functions for monotonic features are monotonic, and the lattices are monotonic with respect to those features, then the ensemble with positive weights on the lattices is monotonic.

## 5 Training the Lattice Ensemble

The key question in training the ensemble is whether to train the base models independently, as is usually done in random forests, or to train the base models jointly, akin to generalized linear models.

### 5.1 Joint Training of the Lattices

In this setting, we optimize (1) jointly over all calibration and lattice parameters, in which case each $\alpha_\ell$ is subsumed by the corresponding base model parameters $\theta_\ell$ and can therefore be ignored. Joint training allows the base models to specialize, and increases the flexibility, but can slow down training and is more prone to overfitting for the same choice of $S$.

### 5.2 Parallelized, Independent Training of the Lattices

We can train the lattices independently in parallel much faster, and then fit the weights $\alpha_t$ in (1) in a second *post-fitting* stage as described in Step 5 below.

**Step 1:** Initialize the calibration function parameters $\nu$ and each tiny lattice's parameters $\theta_\ell$.

**Step 2:** Train the $L$ lattices in parallel; for the $\ell$th lattice, solve the monotonic lattice regression problem [2]:

$$\arg \min_{\theta_\ell} \sum_{i=1}^{n} \mathcal{L}(f(c(x_i[s_\ell]; \nu_\ell); \theta_\ell), y_i) + \lambda R(\theta_\ell), \text{ such that } A\theta \leq 0, \tag{6}$$

where $A\theta \leq 0$ captures the linear inequality constraints needed to enforce monotonicity for whichever features are required, $\mathcal{L}$ is a convex loss function, and $R$ denotes a regularizer on the lattice parameters.

**Step 3:** If separate calibrators are used as per (4) their parameters can be optimized jointly with the lattice parameters in (6). If shared calibrators are used, we hold all lattice parameters fixed and optimize the shared calibration parameters $\nu$:

$$\arg \min_{\nu} \sum_{i=1}^{n} \mathcal{L}(F(x_i ; \theta, \nu, \alpha), y_i), \text{ such that } B\nu \leq 0.$$

$F$ is as defined in (5), and $B\nu \leq 0$ specifies the linear inequality constraints needed to enforce monotonicity of each of the piecewise linear calibration functions.

**Step 4:** Loop on Steps 2 and 3 until convergence.

**Step 5:** Post-fit the weights $\alpha \in \mathbb{R}^L$ over the ensemble:

$$\arg \min_{\alpha} \sum_{i=1}^{n} \mathcal{L}(F(x_i ; \theta, \nu, \alpha), y_i) + \gamma \tilde{R}(\alpha), \text{ such that } \alpha_\ell \geq 0 \text{ for all } \ell, \tag{7}$$

where $\tilde{R}(\alpha)$ is a regularizer on $\alpha$. For example, regularizing $\alpha$ with the $\ell_1$ norm encourages a sparser ensemble, which can be useful for improving evaluation speed. Regularizing $\alpha$ with a ridge regularizer makes the postfit more similar to averaging the base models, reducing variance.

## 6 Fast Evaluation

The proposed lattice ensembles are fast to evaluate. The evaluation complexity of simplex interpolation of a lattice ensemble with $L$ lattices each of size $S$ is $O(LS \log S)$, but in practice one encounters

Table 2: Details for the datasets used in the experiments.

| Dataset | Problem | Features | Monotonic | Train | Validation | Test 1 | Test 2 |
|---|---|---|---|---|---|---|---|
| 1 | Classification | 12 | 4 | 29 307 | 9769 | 9769 | - |
| 2 | Regression | 12 | 10 | 115 977 | - | 31 980 | - |
| 3 | Classification | 54 | 9 | 500 000 | 100 000 | 200 000 | - |
| 4 | Classification | 29 | 21 | 88 715 | 11 071 | 11 150 | 65 372 |

fixed costs, and caching efficiency that depends on the model size. For example, for Experiment 3 with $D = 50$ features, with C++ implementations of both Crystals and random forests both evaluating the base models sequentially, the random forests takes roughly $10\times$ as long to evaluate, and takes roughly $10\times$ as much memory as the Crystals. See Appendix C.5 for further discussions and more timing results.

Evaluating calibration functions can add notably to the overall evaluation time. If evaluating the average calibration function takes time $c$, with shared calibrators the eval time is $O(cD + LS \log S)$ because the $D$ calibrators can be evaluated just once, but with separate calibrators, the eval time is generally worse at $O(LS(c + \log S))$. For practical problems, evaluating separate calibrators may be one-third of the total evaluation time, even when implemented with an efficient binary search. However, if evaluation can be efficiently parallelized with multi-threading, then there is little difference in evaluation time between shared and separate calibrators.

If shared (or no) calibrators are used, merging the ensemble's lattices into fewer larger lattices (see Sec. 2.2) can greatly reduce evaluation time. For example, for a problem $D = 14$ features, we found the accuracy was best if we trained an ensemble of 300 lattices with $S = 9$ features each. The resulting ensemble had $300 \times 2^9 = 153\,600$ parameters. We then merged all 300 lattices into one equivalent $2^{14}$ lattice with only $16\,384$ parameters, which reduced both the memory and the evaluation time by a factor of 10.

# 7   Experiments

We demonstrate the proposals on four datasets. Dataset 1 is the ADULT dataset from the UCI Machine Learning Repository [19], and the other datasets are provided by product groups from Google, with monotonicity constraints for certain features given by the corresponding product group. See Table 2 for details. To efficiently handle the large number of linear inequality constraints ($\sim 100\,000$ constraints for some of these problems) when training a lattice or ensemble of lattices, we used LightTouch [20].

We compared to random forests (RF) [9], an ensemble method that consistently performs well for datasets this size [10]. However, RF makes no attempt to respect the monotonicity constraints, nor is it easy to check if an RF is monotonic. We used a C++ package implementation for RF.

All the hyper parameters were optimized on validation sets. Please see the supplemental for further experimental details.

## 7.1   Experiment 1 - Monotonicity as a Regularizer

In the first experiment, we compare the accuracy of different models in predicting whether income is greater than \$50k for the ADULT dataset (Dataset 1). We compare four models on this dataset: (I) random forest (RF), (II) single unconstrained lattice, (III) single lattice constrained to be monotonic in 4 features, (IV) an ensemble of 50 lattices with 5 features in each lattice and separate calibrators for each lattice, jointly trained. For the constrained models, we set the function to be monotonically increasing in capital-gain, weekly hours of work and education level, and the gender wage gap [21].

Results over 5 runs of each algorithm is shown in Table 3 demonstrate how monotonicity can act as a regularizer to improve the testing accuracy: the monotonic models have lower training accuracy, but higher test accuracy. The ensemble of lattices also improves accuracy over the single lattice, we hypothesize because the small lattices provide useful regularization, while the separate calibrators and ensemble provide helpful flexibility. See the appendix for a more detailed analysis of the results.

Table 3: Accuracy of different models on the ADULT dataset from UCI.

|  | Training Accuracy | Testing Accuracy |
|---|---|---|
| Random Forest | $90.56 \pm 0.03$ | $85.21 \pm 0.01$ |
| Unconstrained Lattice | $86.34 \pm 0.00$ | $84.96 \pm 0.00$ |
| Monotonic Lattice | $86.29 \pm 0.02$ | $85.36 \pm 0.03$ |
| Monotonic Crystals | $86.25 \pm 0.02$ | $\mathbf{85.53 \pm 0.04}$ |

## 7.2 Experiment 2 - Crystals vs. Random Search for Feature Subsets Selection

The second experiment on Dataset 2 is a regression to score the quality of a candidate for a matching problem on a scale of $[0, 4]$. The training set consists of $115\,977$ past examples, and the testing set consists of $31\,980$ more recent examples (thus the samples are not IID). There are $D = 12$ features, of which $10$ are constrained to be monotonic on all the models compared for this experiment. We use this problem with its small number of features to illustrate the effect of the feature subset choice, comparing to RTLs optimized over $K = 10\,000$ different trained ensembles, each with different random feature subsets.

All ensembles were restricted to $S = 2$ features per base model, so there were only 66 distinct feature subsets possible, and thus for a $L = 8$ lattice ensemble, there were $\binom{66}{8} \simeq 5.7 \times 10^9$ possible feature subsets. Ten-fold cross-validation was used to select an RTL out of 1, 10, 100, 1000, or $10\,000$ RTLs whose feature subsets were randomized with different random seeds. We compared to a calibrated linear model and a calibrated single lattice on all $D = 12$ features. See the supplemental for further experimental details.

Figure 2 shows the normalized mean squared error (MSE divided by label variance, which is 0 for the oracle model, and 1 for the best constant model). Results show that Crystals (orange line) is substantially better than a random draw of the feature subsets (light blue line), and for mid-sized ensembles (e.g. $L = 32$ lattices), Crystals can provide very large computational savings ($1000\times$) over the RTL strategy of randomly considering different feature subsets.

## 7.3 Experiment 3 - Larger-Scale Classification: Crystals vs. Random Forest

The third experiment on Dataset 3 is to classify whether a candidate result is a good match to a user. There are $D = 54$ features, of which $9$ were constrained to be monotonic. We split 800k labelled samples based on time, using the 500k oldest samples for a training set, the next 100k samples for a validation set, and the most recent 200k samples for a testing set (so the three datasets are not IID).

Results over 5 runs of each algorithm in Figure 3 show that Crystals ensemble is about $0.25\%$ - $0.30\%$ more accurate on the testing set over a broad range of ensemble sizes. The best RF on the validation set used 350 trees with a leaf size of 1 and the best Crystals model used 350 lattices with 6 features per lattice. Because the RF hyperparameter validation chose to use a minimum leaf size of 1,

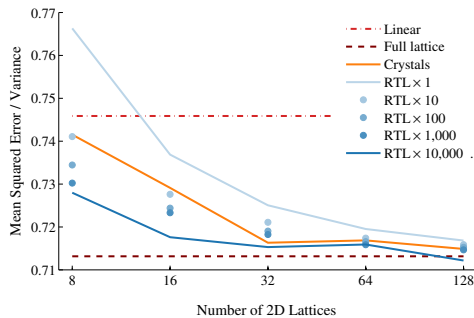

Figure 2: Comparison of normalized mean squared test error on Dataset 2. Average standard error is less than $10^{-4}$.

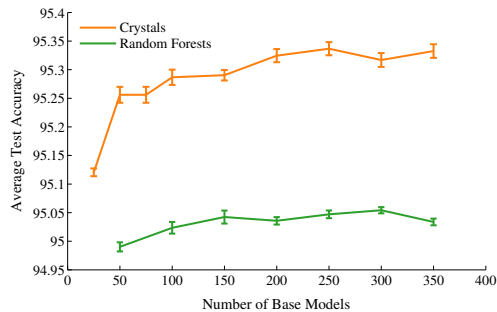

Figure 3: Test accuracy on Dataset 3 over the number of base models (trees or lattices). Error bars are standard errors.

Table 4: Results on Dataset 4. Test Set 1 has the same distribution as the training set. Test Set 2 is a more realistic test of the task. Left: Crystals vs. random forests. Right: Comparison of different optimization algorithms.

| | Test Set 1 Accuracy | Test Set 2 Accuracy |
|---|---|---|
| Random Forest | $75.23 \pm 0.06$ | $90.51 \pm 0.01$ |
| Crystals | $75.18 \pm 0.05$ | $\mathbf{91.15} \pm 0.05$ |

| Lattices Training | Calibrators Per Lattice | Test Set 1 Accuracy |
|---|---|---|
| Joint | Separate | $\mathbf{74.78} \pm 0.04$ |
| Independent | Separate | $72.80 \pm 0.04$ |
| Joint | Shared | $74.48 \pm 0.04$ |

the size of the RF model for this dataset scaled linearly with the dataset size, to about 1GB. The large model size severely affects memory caching, and combined with the deep trees in the RF ensemble, makes both training and evaluation an order of magnitude slower than for Crystals.

### 7.4 Experiment 4 - Comparison of Optimization Algorithms

The fourth experiment on Dataset 4 is to classify whether a specific visual element should be shown on a webpage. There are $D = 29$ features, of which 21 were constrained to be monotonic. The Train Set, Validation Set, and Test Set 1 were randomly split from one set of examples, whose sampling distribution was skewed to sample mostly difficult examples. In contrast, Test Set 2 was uniformly randomly sampled, with samples from a larger set of countries, making it a more accurate measure of the expected accuracy in practice.

After optimizing the hyperparameters on the validation set, we independently trained 10 RF and 10 Crystal models, and report the mean test accuracy and standard error on the two test sets in Table 4. On Test Set 1, which was split off from the same set of difficult examples as the Train Set and Validation Set, the random forests and Crystals perform statistically similar. On Test Set 2, which is six times larger and was sampled uniformly and from a broader set of countries, the Crystals are statistically significantly better. We believe this demonstrates that the imposed monotonicity constraints effectively act as regularizers and help the lattice ensemble generalize better to parts of the feature space that were sparser in the training data.

In a second experiment with Dataset 4, we used RTLs to illustrate the effects of shared vs. separate calibrators, and training the lattices jointly vs. independently.

We first constructed an RTL model with 500 lattices of 5 features each and a separate set of calibrators for each lattice, and trained the lattices jointly as a single model. We then separately modified this model in two different ways: (1) training the lattices independently of each other and then learning an optimal linear combination of their predictions, and (2) using a single, shared set of calibrators for all lattices. All models were trained using logistic loss, mini-batch size of 100, and 200 loops. For each model, we chose the optimization algorithms' step sizes by finding the power of 2 that maximized accuracy on the validation set.

Table 4 (right) shows that joint training and separate calibrators for the different lattices can provide a notable and statistically significantly increase in accuracy, due to the greater flexibility.

## 8 Conclusions

The use of machine learning has become increasingly popular in practice. That has come with a greater demand for machine learning that matches the intuitions of domain experts. Complex models, even when highly accurate, may not be accepted by users who worry the model may not generalize well to new samples.

Monotonicity guarantees can provide an important sense of control and understanding on learned functions. In this paper, we showed how ensembles can be used to learn the largest and most complicated monotonic functions to date. We proposed a measure of pairwise feature interactions that can identify good feature subsets in a fraction of the computation needed for random feature selection. On real-world problems, we showed these monotonic ensembles provide similar or better accuracy, and faster evaluation time compared to random forests, which do not provide monotonicity guarantees.

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
