[Supplementary Material · ensemble_nips_appendix.pdf]

# Fast and Flexible Monotonic Functions with Ensembles of Lattices
# Appendix

**K. Canini, A. Cotter, M. R. Gupta, M. Milani Fard, J. Pfeifer**
Google Inc.
1600 Amphitheatre Parkway, Mountain View, CA 94043
{canini,acotter,mayagupta,mmilanifard,janpf}@google.com

This appendix contains three sections: Formulas for multilinear and simplex interpolation are given in Section A, proof of the theorem is included in B and details and extensions of the experimental analysis is discussed in Section C.

Table 1: Key notation used in the paper.

| Symbol | Definition |
|---|---|
| $D$ | number of features |
| $\mathcal{D}$ | set of features $1, 2, \ldots, D$ |
| $S$ | number of features in each lattice |
| $L$ | number of lattices in the ensemble |
| $s_\ell \subset \mathcal{D}$ | $\ell$th lattice's set of $S$ feature indices |
| $(x_i, y_i) \in [0, 1]^D \times \mathbb{R}$ | $i$th training example |
| $x[s_\ell] \in [0, 1]^S$ | $x$ for feature subset $s_\ell$ |
| $\Phi(x) : [0, 1]^D \to [0, 1]^{2^D}$ | linear interpolation weights for $x$ |
| $v \in \{0, 1\}^D$ | a vertex of a $2^D$ lattice |
| $\theta \in \mathbb{R}^{2^D}$ | parameter values for a $2^D$ lattice |

## A   Review of Multilinear and Simplex Interpolation

Let the matrix $V$ have as its rows the $2^D$ vertices of a unit hypercube, so that each row is some $v \in \{0, 1\}^D$. Let $\theta \in \mathbb{R}^{2^D}$ be values defined for each vertex of the unit hypercube. Let $x \in [0, 1]^D$. Then $f(x) = \theta^T \Phi(x)$ is a linear interpolation of $\theta$ at $x$ if the weight function $\Phi(x)$ satisfies the following system of equations for all $x$:

$$V^T \Phi(x) = x \qquad (1)$$

$$\mathbf{1}^T \Phi(x) = 1 \qquad (2)$$

$$\Phi(x) \geq \mathbf{0}. \qquad (3)$$

We review two formulas for $\Phi$, the common *multilinear* interpolation, and the more efficient *simplex* interpolation (also known as the Lovasz extension when used to relax submodular functions for optimization [1]). For illustrations and more details, see Gupta et al [2].

### A.1   Multilinear Interpolation

Multilinear is the multi-dimensional generalization of the familiar bilinear interpolation often used to upsample images. Multilinear interpolation weights all $2^D$ vertices of a unit hypercube, producing

a smooth continuous interpolation over the unit hypercube. The multilinear interpolation weights $\Phi : [0,1]^D \rightarrow [0,1]^{2^D}$ can be expressed component-wise in terms of $\phi : [0,1]^D \times \{0,1\}^D \rightarrow [0,1]$, where each component $\phi(x,v)$ is the interpolation weight for an input $x$ on a particular hypercube vertex $v \in \{0,1\}^D$:

$$\phi(x,v) = \prod_{d=1}^{D} x[d]^{v[d]}(1-x[d])^{1-v[d]}. \tag{4}$$

That is, each interpolation weight multiplies-in either $x[d]$ or $1 - x[d]$ for each of the $D$ components.

### A.2 Simplex Interpolation

Simplex interpolation implicitly divides the unit hypercube into $D!$ simplices, each with $D+1$ vertices, and fits a linear hyperplane to each simplex, producing a continuous locally linear surface. Let $\pi$ be a permutation of the first $D$ natural numbers that sorts the components of $x$ such that $x[\pi[1]] \leq x[\pi[2]] \cdots \leq x[\pi[D]]$. Let $v_d \in \{0,1\}^D$ be 1 for each component $\pi[1], \pi[2], \ldots, \pi[d-1]$ and 0 otherwise, for $d \in \{1, \ldots, D+1\}$ (all elements of $v_1$ are zero, and of $v_{D+1}$ are one, with each $v_d$ containing $d-1$ nonzeros). The simplex interpolation weights $\Phi(x)$ can be expressed, as above, in terms of the components $\phi(x,v)$, with:

$$\begin{aligned} \phi(x,v_1) &= 1 - x[\pi[1]], & (5) \\ \phi(x,v_d) &= x[\pi[d-1]] - x[\pi[d]] \ \ \text{for } d = 2, \ldots, D, \\ \phi(x,v_{D+1}) &= x[D], \\ \phi(x,v) &= 0 \ \ \text{otherwise}. \end{aligned}$$

The main advantage of simplex interpolation is that all but $D+1$ of the interpolation weights are zero, with the consequence that, in order to evaluate the value of the lattice for a particular input $x$, one need only check $D+1$ of the lattice parameters. Furthermore, the indices of these relevant parameters may be found in only $O(D \ln D)$ time (due to the sorting).

## B   Proof of Theorem

Figure 1 illustrates some aspects of the theorem, proven below.

**Proof of Theorem** First, Lemma 1 (below) establishes that interpolating a lattice $(s_1, \theta_1)$ is equivalent to interpolating a lattice on a superset of features $(s_1 \cup s_2, \tilde{\theta}_1)$ and defines the relationship between $\tilde{\theta}_1$ and $\theta_1$.

Applying Lemma 1 to two lattices $(s_1, \theta_1)$ and $(s_2, \theta_2)$,

$$\begin{aligned} \theta_1^T \Phi(x[s_1]) &+ \theta_2^T \Phi(x[s_2]) \\ &= \tilde{\theta}_1^T \Phi(x[s_1 \cup s_2]) + \tilde{\theta}_2^T \Phi(x[s_1 \cup s_2]) \\ &= (\tilde{\theta}_1 + \tilde{\theta}_2)^T \Phi(x[s_1 \cup s_2]) \\ &= \theta^T \Phi(x[s_1 \cup s_2]). \end{aligned}$$

This establishes the result for a pair of lattices where the merged lattice parameter vector $\theta = (\tilde{\theta}_1 + \tilde{\theta}_2)$. This pair-result can be applied inductively to the whole ensemble, proving the theorem.

**Lemma 1:** Denote the vertices of the lattice $(s, \theta)$ as $v \in \{0,1\}^S$, and the vertices of the lattice $(\mathcal{D}, \tilde{\theta})$ as $\tilde{v} \in \{0,1\}^D$. Given a lattice $(s, \theta)$ with $s \in \mathcal{D}$, construct the lattice $(\mathcal{D}, \tilde{\theta})$ with $\tilde{\theta}(\tilde{v}) = \theta(v)$ for any $\tilde{v}$ and $v$ such that $\tilde{v}[s] = v$. Then,

$$\theta^T \Phi(x[s]) = \tilde{\theta}^T \Phi(x), \tag{6}$$

for either the multilinear or simplex interpolation weights $\Phi$.

Figure 1: Illustration of the theorem: A lattice's parameters can be copied across an additional feature to create an equivalent lattice defined on a superset of features of the original lattice. For example, a lattice on features 1 and 2 can be expressed as a lattice on features 1,2, and 3, as shown. By linearity, two lattices defined on the same feature set can be averaged to form an equivalent lattice on that feature set. Combining these steps, an ensemble of lattices can be merged into an equivalent lattice defined on the union of features used in the base models.).

**Proof of Lemma 1:** Write the interpolation of the larger lattice $(\mathcal{D}, \tilde{\theta})$ as a sum over its unit hypercube vertices:

$$\tilde{\theta}^T \Phi(x) = \sum_{\tilde{v} \in \{0,1\}^D} \tilde{\theta}(\tilde{v}) \phi(x, \tilde{v})$$

$$= \sum_{v \in \{0,1\}^S} \sum_{\tilde{v} \text{ s.t. } \tilde{v}[s] = v} \tilde{\theta}(\tilde{v}) \phi(x, \tilde{v}) \tag{7}$$

(by grouping terms of the sum above)

$$= \sum_{v \in \{0,1\}^S} \theta(v) \left( \sum_{\tilde{v}[s]=v} \phi(x, \tilde{v}) \right) \tag{8}$$

(because $\tilde{\theta}(\tilde{v}) = \theta(v)$ for $\tilde{v}[s] = v$).

$$= \theta^T \Phi(x[s]) \text{ if } \sum_{\tilde{v}[s]=v} \phi(x, \tilde{v}) = \phi(x[s], v). \tag{9}$$

We first show the condition on line (9) holds for multilinear interpolation, then for simplex interpolation.

For multilinear interpolation, we first prove the special case that $D = S + 1$, and without loss of generality, that the subset features are $s = 1, 2, \cdots, D - 1$. Then there are only two vertices that satisfy $\tilde{v}[s] = v$, one with additional component 0, and one with additional component 1. Thus:

$$\sum_{\tilde{v}[s]=v} \phi(x, \tilde{v}) = \phi(x, [v, 0]) + \phi(x, [v, 1])$$

then by (4)
$$= \phi(x[s], v)(1 - x[D]) + \phi(x[s], v)(x[D])$$
$$= \phi(x, v).$$

The general case follows by induction.

For simplex interpolation, we will show that the $D+1$ non-zero simplex interpolation weights $\phi(x,\tilde{v})$ can be partitioned, with each partitioning summing to one of the $S+1$ non-zero simplex interpolation weights $\phi(x,v)$.

Let $\pi[s[k]]$ denote the index in $x$ of the $k$th largest value in the feature subset $x[s]$. Then we consider the three non-zero cases of (5) one-by-one. First, by (5), the simplex weight on $v_1$ is:

$$\phi(x[s],v_1) = 1 - x[\pi[s[1]]],$$
$$\text{then expand this sum with terms that cancel:}$$
$$= 1 + (-x[\pi[1]] + x[\pi[1]]) +$$
$$\ldots + (-x[\pi[s[1]-1]] + x[\pi[s[1]-1]]) - x[\pi[s[1]]],$$
$$= (1 - x[\pi[1]]) + x[\pi[1]] +$$
$$\ldots - x[\pi[s[1]-1]] + (x[\pi[s[1]-1]] - x[\pi[s[1]]]),$$
$$\text{then group pairs of terms to form}$$
$$= \phi(x,\tilde{v}_1) + \phi(x,\tilde{v}_2) + \ldots + \phi(x,\tilde{v}_{\pi[s[1]]})$$

$$\tag{10}$$

$$= \sum_{\tilde{v}[s]=v_1} \phi(x,\tilde{v}),$$

where the last line holds because $v_1$ is a vector of $S$ zeros by definition, and to be included in (10) all of the terms must also have zeros for the components of $s$, and any other vertex in the full lattice with zeros for the components of $s$ must have zero interpolation weight by (5).

Consider next the simplex weight on $v_S$. Recall that $x[\pi[s[S]]]$ denotes the largest component of $x$ that is also in $s$, and thus

$$\phi(x[s],v_S) = x[\pi[s[S]]],$$
$$\text{then expand this sum with terms that cancel:}$$
$$= x[\pi[s[S]]] - x[\pi[s[S]+1]] + x[\pi[s[S]+1]]$$
$$+ \ldots - x[\pi[D]] + x[\pi[D]],$$
$$\text{then by (5),}$$
$$= \phi(x,\tilde{v}_{\pi[s[S]]}) + \ldots + \phi(x,\tilde{v}_{D-1}) + \phi(x,\tilde{v}_D) \tag{11}$$
$$= \sum_{\tilde{v}[s]=v_S} \phi(x,\tilde{v}),$$

where the last line holds because $v_S$ is a vector of ones by definition, and all the terms in (11) must also be all-ones for the $s$ components, and any other vertex in the full lattice with ones for the components of $s$ must have zero interpolation weight by (5).

The last cases to consider are the simplex weights for $v_k$, where $k = 2, \cdots, S-1$, from (5):

$$\phi(x[s],v_k) = x[\pi[s[k-1]]] - x[\pi[s[k]]],$$
$$\text{then expand this sum with terms that cancel:}$$
$$= x[\pi[s[k-1]]] \tag{12}$$
$$- x[\pi[s[k-1]+1]] + x[\pi[s[k-1]+1]]$$
$$\ldots - x[\pi[s[k]-1]] + x[\pi[s[k]-1]]$$
$$- x[\pi[s[k]]],$$
$$\text{then by (5)}$$
$$= \phi(x,\tilde{v}_{\pi[s[k-1]+1]}) + \phi(x,\tilde{v}_{\pi[s[k-1]+2]})$$
$$+ \ldots + \phi(x,\tilde{v}_{\pi[s[k]]}) \tag{13}$$
$$= \sum_{\tilde{v}[s]=v_k} \phi(x,\tilde{v}),$$

where the last line holds similarly to the two cases above.

# C Details of Experimental Analysis

This section includes further details and discussions on the experiments presented in the paper.

## C.1 Experiment 1: Monotonicity as a Regularizer

For RF, bagging ratio choices were $\{0.2, 0.4, 0.6, 0.8\}$, the number of features per split choices were $\{3, 5, 7, 9\}$, the number of samples in a leaf choices were $\{1, 4, 16, 64\}$, and the choices for the number of trees were $\{50, 100, 150, 200, 250, 300, 350\}$ with the optimal 0.2, 5, 1 and 250.

Lattice and Crystals models had these hyperparameter combos: calibration step size $\{0.01, 0.02, 0.04, 0.08, 0.16\}$, and lattice step size $\{0.1, 0.2, 0.4, 0.8, 1.6\}$. The optimal values were $0.04$ and $0.8$ for unconstrained lattice, $0.16$ and $0.8$ for monotonic lattice, and $0.04$ and $0.4$ for the Crystals model. We used mini-batch size of 100 and constraint distribution size of 100. Each model was trained for 50 loops.

Figure 2 shows the partial dependence plot on two key features: age and education. Each point on the plot is calculated by replacing the age and education features in the test set with the plot x and y values, then averaging over the model output of the resulting "hallucinated" test set to get the z value. Education is a numerical feature (see the dataset description on UCI) increasing in value with higher education level.

The Crystals ensemble model is monotonic in the ensemble feature whereas the random forest model is not constrained. The Crystals model has a smoother surface compare to the random forest, and is clearly using the monotonicity constraint as useful domain knowledge. The random forest model does not seem to be using the education feature by much. The plot resulting from the ensemble model is easy to interpret: income tends to grow with age until the retirement, but the effect is much larger for the highly educated.

Figure 2: Partial dependence of the trained model projected on two features.

## C.2 Experiment 2: Feature Selection, Crystals vs. Randomized Search

Ensembles using more 2D lattices performed better. Training 10000 different RTL ensembles and then choosing the one with the best cross-validation performance resulted in the best testing performance for all ensemble sizes. For small ensembles with only 8 or 16 base models, Crystals performed as well as choosing between 10 RTLs. For the mid-size ensemble with 32 feature pairs, Crystals performed better than cross-validating amongst 1000 RTLs. Since the Crystals training time is similar to that of a single RTL ensemble, this was a $1000\times$ computational savings. However separate RTLs can be trained in parallel to significantly reduce the gap.

## C.3 Experiment 3: Large-scale Classification, Crystals vs. Random Forest

For RF, bagging ratio choices were $\{0.2, 0.4, 0.6, 0.8\}$, the number of features per split choices were $\{5, 10, 15, 20\}$, the number of samples in a leaf choices were $\{1, 4, 16, 64\}$, for a total of 64 hyperparameter combos. Crystals also had 64 hyperparameter combos: the number of features per lattice

$\{4, 6, 8, 10\}$, the step size $\{0.25, 0.5, 1, 2\}$, and the mini-batch size for SGD $\{250, 500, 1000, 2000\}$. Crystals training was fixed at 200 loops through the training samples. All the datasets have a biased distribution of $90\%$ negative examples and $10\%$ positive examples. The final threshold on all models was chosen so that each model predicted $10\%$ of the testing set to be positive.

The best RF on the validation set had $95.21\%$ validation accuracy and used 350 trees, tried 15 features per split, had a bagging fraction of 0.6, and minimum leaf size of 1. The best Crystals model on the validation set had $95.53\%$ validation accuracy and used 350 lattices, 6 features per lattice, mini-batch size of 250, and step size of 1.

### C.4  Experiment 4: Large-scale Classification, Crystals vs. Random Forest

Based on preliminary experiments, we validated RF's hyperparameters from: bagging ratio $\{0.5, 0.6, 0.7, 0.8, 0.9\}$, number of features for split $\{3, 5, 7, 9\}$, number of samples in a leaf $\{1, 4, 16, 64\}$, and number of trees $\{50, 100, 150, 200, 250, 300, 350\}$, and the best RF on the validation set used hyperparameters: $0.8, 5, 1, 250$ (respectively). For Crystals, we used squared loss and fixed the number of features per lattice at 8. We validated the model over step sizes $\{0.1, 0.2, 0.3, 0.4, 0.5\}$, mini-batch sizes $\{40, 60, 80, 100\}$, and number of lattices $\{50, 100, 150, 200, 250\}$, and the best Crystals on the validation set used hyperparameters: $0.4, 80, 150$ respectively.

### C.5  Experiments on Model Evaluation Time

Figure 3 compares the evaluation time of ensembles of 100 or 10 lattices on a single-threaded 3.5GHz Intel Ivy Bridge processor, for different numbers of subset features $S$. The plot shows that 10 tiny lattices each with $D = 20$ features (right extreme) is faster to evaluate than 100 lattices each with $D = 2$ features (left extreme).

Figure 3: Plot shows benchmarked evaluation times for simplex interpolation of an ensemble with either 100 or 10 tiny lattices, where each tiny lattice has the number of features $S$ shown on the x-axis and $2^S$ parameters.