[Reviews · NeurIPS 2016]

Reviewer 1

Summary

The paper deals with learning monotonic functions. This can be useful for the data, for which there are some inputs that are known to be positively (or negatively) related to the output, and in such cases training the model to respect that monotonic relationship can provide regularization, and makes the model more interpretable. The main problem which is dealt in the paper is that flexible monotonic functions are computationally challenging to learn beyond a few features. The paper proposes to learn ensembles of monotonic calibrated interpolated look-up tables (lattices). A key contribution is an automated algorithm for selecting feature subsets for the ensemble base models. The algorithm is experimentally compared against random forests, and is shown to produce similar or better accuracy, while providing guaranteed monotonicity consistent with prior knowledge, smaller model size and faster evaluation.

Qualitative Assessment

The paper is heavily based on the previous work on lattice (and monotone lattice) regression, and is thus not self-contained. It is not even explained in the main part of the paper what is the actual function which a lattice represents (two examples are found in the supplementary materials, but I still found the explanation two brief). I think this should definitely be a part of the main paper, otherwise the reader is forced to check the previous work on this topic to understand what kind model is actually being considered. The paper barely touches upon the computational complexity issues of optimizing (3), (7), and unnamed eq. in Step 3. While (7) is convex given calibrators being fixed, it is not clear whether it is convex when jointly optimized over calibration and lattice parameters (I doubt it is). Moreover, (3) seems highly complex and combinatorial optimization criterion. The optimization complexity of eq. is Step 3 is unclear to me (note that Step 3 requires knowing ensemble weights \alpha, which are only computed in Step 5; how is this issue solved?). The authors should clearly state which problems are easy to optimize, and which are only solved heuristically, likely returning suboptimal solution. Given a complex training of lattice ensemble I missed the computational times of training. Since the authors report faster evaluation of their method over random forests at the testing phase, they should also report training time comparing to the competitors. The experimental results are not very convincing: only around 1% of accuracy improvement is reported comparing to the random forests; and it is not clear if random forest is a good fit for these data sets. It also does not help that all data sets except the first one are proprietary (hard to find other experimental studies on these data sets). It would be beneficial to check at least one more state-of-the-art flexible regression method, such as, e.g., gradient boosting, on these data sets (the unconstrained lattice, as another competitor, is not enough, as it is not particularly known algorithm). For instance, xgboost (an implementation of gradient boosting) easily achieves over 86% accuracy on Adult, see https://jessesw.com/XG-Boost/. RuleFit algorithm of Friedman & Popescu (http://statweb.stanford.edu/~jhf/ftp/RuleFit.pdf) gives 13.6% test error (86.4% accuracy), etc. Apparently, they all beat both random forests and monotone lattices Adding more methods to the comparison would help to resolve the question, whether exploiting monotonicity actually gives an advantage in predictive performance. Please explain what is novel in the paper comparing to Ref. [2]. Both papers share monotonic lattice regression, feature calibration and some of the optimization criteria (e.g., for joint optimization of calibration and lattice parameters). It looks as if the novel part is the ensemble learning of lattices, together with crystals algorithm to select features for lattice construction. And the parts related to incorporating monotonicity are not.

Confidence in this Review

2-Confident (read it all; understood it all reasonably well)


Reviewer 2

Summary

The article shows a way to build ensembles of lattice look-up tables that code monotonic relations with respect to the output. The creation of an ensembles of simpler lattices, instead of a more complex single lattice, is what makes the proposed algorithm computationally feasible. The article is tested on 4 datasets (1 public and 3 undisclosed) showing a slight improvement over Random Forest in accuracy and large improvement in computational time required to train and to test the models.

Qualitative Assessment

The article is interesting and promising. However, I think the experimental section is weak and in my opinion the article puts too much information in the supplemental material. The main concerns I have with this article are: * There is a very related article that is not cited nor discussed. This is: Wojciech Kotlowski and Roman Slowinski, Rule Learning with Monotonicity Constraints, ICML 2009. "The algorithm first monotonizes the data using a nonparametric classification procedure and then generates a rule ensemble consistent with the training set" The authors should discuss this reference and stress in what way the current proposal improves over Kotlowski and Slowinski's. In my opinion this should be done by also including this algorithm in the experimental comparison. * It is not clear how many realizations are carried out for each experiment. In experiment 1 nothing is said but results are given with standard deviation. In the temporal experiment it would be better, instead of reporting a single experiment result, to do a sliding window experiment, to have more results. The experiment 4 it is not clear if the four subsets (train, validation, test 1 and 2) are disjoin or not. Is the same train+val used to train the models that are tested in 1 and 2? The following sentence is not clear either: "After optimizing the hyperparameters in the validation set we independently trained 10 RF and 10 crystal models..." Did you use one single validation set and 10 different training set? was it done with cross validation? * Every experiment optimizes a different set of parameters and in a different range of values. The same parameter grid search should be used for all experiments. * In addition, Random Forest (RF) does not really need any optimization to get to good results. Better results should be obtained using both train+val for training using default values. In any case, out-of-bag optimization could be used with RF to obtain the best parameter set without reducing the data for training. * This is not very important but in pg 7 it is said "Because RF ...chose ...minimum leaf size 1, the size of RF....scaled linearly with the dataset size...". This could be true in this case but this is not necessary true for all classification tasks as 1 is only the minimum. A log size increase could also be observed or even no increase at all. * Finally, it is claimed that the learned models are more interpretable. This might be true for single lattices. However, the article does not explain how this interpretation could be done when building ensembles of hundreds of lattices.

Confidence in this Review

2-Confident (read it all; understood it all reasonably well)


Reviewer 3

Summary

This paper builds on recent work on monotonic calibrated interpolated look-up tables and introduces ensembles of these tables. Much like the random forests, these ensembles do not change the expressiveness of the base model class (except if feature calibration is done separately on each model). Still, they provide a powerful regularisation, because each individual interpolated look-up table has only a few features. In the experiments conducted on 1 public and 3 private industrial datasets the proposed method is faster and more accurate than random forests.

Qualitative Assessment

This is very, very interesting work and enjoyed reading this very much. Of course, partly my admiration is due to my first encounter with monotonic calibrated interpolated look-up tables, presented in the soon-to-appear paper [2] in JMLR, currently available from Arxiv. Still, the step to building ensembles is a non-trivial and significant improvement, resulting in better and more interpretable models than random forests. Importantly, the models are more reliable as well, because their performance in sparsely populated feature space regions is very intuitive and meaningful. Certainly it would have been even better if the authors could have presented experiments on more public data, as well as make the code available. My other few criticisms are very minor, just to improve presentation of this paper. I first got the impression that the paper does not present enough details about the method in paper [2], forcing the reader to read [2] first. Actually, the supplementary provides almost enough detail about [2]. The only thing missing is a pointer to section A of the supplementary material early on in the Section 2. But in addition to that, I think that the very start of Section 2, lines 43 to 54 would benefit from a bit more explanation. Especially, the first mentioning of linearly-interpolated look-up table begs some explanation. In Section 7.1 I think it is worth emphasizing once (maybe in the caption of Table 2?) more that the values of parameters are given in the supplementary. I found typos on lines 77 (interpolaion), 110 (define->defined), 111 (torsion->torsion is), 211 (the weather->weather).

Confidence in this Review

2-Confident (read it all; understood it all reasonably well)


Reviewer 4

Summary

The authors introduce a new learning ensembles of monotonic functions. They propose an automatic way of selecting features for the lattices of the ensemble. The key point is to ensure monoticity in predictors in order to ease (and consolidate) the interpretation of the results. The authors state the method and describe the feature selection procedure and the way they trained them, before leading experiments on 4 datasets.

Qualitative Assessment

The paper is well written but it is sometimes difficult to follow (especially section 2 and 3). This may be due by the fact that several notations are introduced a few lines after their first use, which renders difficult the understanding. Furthermore, it may seem obvious to the authors (and lots of other people I admit), but the term "look-up table" is never defined, and it is not clear to me what this is. Reference [2] is very often pointed out, hence the paper does not appear self-contained enough (at least I have great trouble to understand the current paper). About the current method, I did not understand how S is chosen, even if it appears to be a very important parameter of the algorithm.

Confidence in this Review

1-Less confident (might not have understood significant parts)


Reviewer 5

Summary

The authors have proposed an fast and flexiable montic function learning method by ensemble of lattice. I guess it is an extension of the previous work shown in reference [2].

Qualitative Assessment

The authors proposd to ensemble several lattices to learn monotonic functions. I am not an expert in this field and the paper seems interesting. (1) There are plenty of theoretical and experimental results in this paper. Nevertheless, the paper is very hard to follow, although I have read this paper for several times and tried to catch the main idea. I cannot totally understand how to learn monotonic functions with ensembles of lattices. (2) I think the most related work is monotic lattice regression. What is the essencial different between your work and the previous one? Adding felxiablity and computational efficiency? (3) For clarity, it is preferable to listing an algorithmic table.

Confidence in this Review

1-Less confident (might not have understood significant parts)


Reviewer 6

Summary

The paper describes a method for monotonic regression. The main idea is to represent the regression function as a sum of "tiny" lattices, each depending on a small number of the variables. The authors discuss several methods for selecting the variables in each lattice, including random selection, and the "crystals" algorithm.

Qualitative Assessment

This is a very clear paper addressing a moderately interesting problem.

Confidence in this Review

1-Less confident (might not have understood significant parts)


Reviewer 7

Summary

This paper introduces an algorithm that learns a flexible monotonic ensemble for classification and regression problems. Monotonicity helps the user to understand the prediction model better and increases the trust in the model. Previous algorithms did not scale well with an increasing number of features. The proposed ensemble algorithm can be learned efficiently and performs well compared to random forest on large classification an regression problems. A second contribution of this paper is an algorithm to select feature subsets which show substantial nonlinear interactions.

Qualitative Assessment

This paper is well written and the results are promising. The introduced method is the first algorithm with monotonicity guarantees that scales well with more than a few features. The ensemble of lattices performs as good or even better than random forest. The crystal algorithm seems to be an effective way to find features subsets with substantial non-linear interactions. Is the crystal method itself a meaningful contribution that can inform other ensemble methods? For instance, could the size of a random forest be reduced by selecting feature subsets before training? I am wondering why each experiment is only conducted on one of the data sets. Do the results of the experiments hold for all data sets? In general, I believe that a bigger test-bed of public data sets would strengthen this paper. All experiments are based on fairly large data sets. I am wondering how the method performs on small or medium sized standard UCI data sets. How sensitive is the method for the choice of the hyperparameters (step size, features per lattice)? We know for instance, that random forests often do very well with the default parameter values so that a validation set is not even necessary.

Confidence in this Review

1-Less confident (might not have understood significant parts)